# Neural Assets: 3D-Aware Multi-Object Scene Synthesis with Image Diffusion Models

**Ziyi Wu**[3,4,†], **Yulia Rubanova**[1], **Rishabh Kabra**[1,5], **Drew A. Hudson**[1],
**Igor Gilitschenski**[3,4], **Yusuf Aytar**[1], **Sjoerd van Steenkiste**[2],
**Kelsey R. Allen**[1], **Thomas Kipf**[1]

[1]Google DeepMind  [2]Google Research  [3]University of Toronto  [4]Vector Institute  [5]UCL

## Abstract

We address the problem of multi-object 3D pose control in image diffusion models. Instead of conditioning on a sequence of text tokens, we propose to use a set of per-object representations, *Neural Assets*, to control the 3D pose of individual objects in a scene. Neural Assets are obtained by pooling visual representations of objects from a reference image, such as a frame in a video, and are trained to reconstruct the respective objects in a different image, e.g., a later frame in the video. Importantly, we encode object visuals from the reference image while conditioning on object poses from the target frame. This enables learning disentangled appearance and pose features. Combining visual and 3D pose representations in a sequence-of-tokens format allows us to keep the text-to-image architecture of existing models, with Neural Assets in place of text tokens. By fine-tuning a pre-trained text-to-image diffusion model with this information, our approach enables fine-grained 3D pose and placement control of individual objects in a scene. We further demonstrate that Neural Assets can be transferred and recomposed across different scenes. Our model achieves state-of-the-art multi-object editing results on both synthetic 3D scene datasets, as well as two real-world video datasets (Objectron, Waymo Open). Additional details and video results are available at our [project page](#).

## 1 Introduction

From animation movies to video games, the field of computer graphics has long relied on a traditional workflow for creating and manipulating visual content. This approach involves the creation of 3D assets, which are then placed in a scene and animated to achieve the desired visual effects. With the recent advance of deep generative models [26, 50, 77, 82], a new paradigm is emerging. Diffusion models have achieved promising results in content creation [22, 44, 70, 79] by training on web-scale text-image data [85]. Users can now expect realistic image generation, depicting almost everything describable in text. However, text alone is often insufficient for precise control over the output image.

To address this challenge, an emerging body of work has investigated alternative ways to control the image generation process. One line of work studies different forms of conditioning inputs, such as depth maps, surface normals, and semantic layouts [59, 103, 116]. Another direction is personalized image generation [30, 58, 81], which aims to synthesize a new image while preserving particular aspects of a reference image (e.g., placing an object of interest on a desired background). However, these approaches are still fundamentally limited in their 3D understanding of objects. As a result, they cannot achieve intuitive object control in the 3D space, e.g., rotation. While some recent works introduce 3D geometry to the generation process [8, 61, 67], they cannot handle multi-object real-world scenes as it is hard to obtain scalable training data (paired images and 3D annotations).

---

[†]Work done while interning at Google.

Contact: `tkipf@google.com`. Project page: `neural-assets.github.io`

38th Conference on Neural Information Processing Systems (NeurIPS 2024).

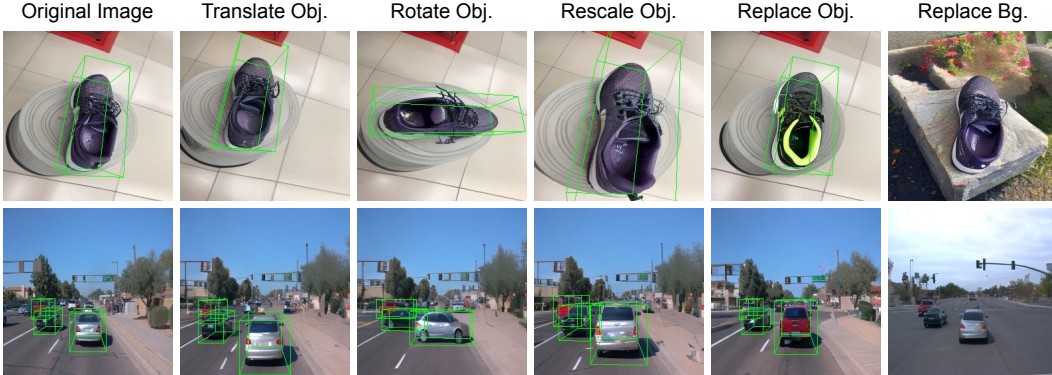

| Original Image | Translate Obj. | Rotate Obj. | Rescale Obj. | Replace Obj. | Replace Bg. |

Figure 1: **3D-aware editing with our Neural Asset representations**. Given a source image and object 3D bounding boxes, we can translate, rotate, and rescale the object. In addition, we support compositional generation by transferring objects or backgrounds across images.

We address these limitations by taking inspiration from cognitive science to propose a scalable solution to 3D-aware multi-object control. When humans move through the world, their motor systems keep track of their movements through an efference copy and proprioceptive feedback [4, 96]. This allows the human perceptual system to track objects accurately across time even when the object's relative pose to the observer changes [28]. We use this observation to propose the use of *videos* of multiple objects as a scalable source of training data for 3D multi-object control. Specifically, for any two frames sampled from a video, the naturally occurring changes in the 3D pose (e.g., 3D bounding boxes) of objects can be treated as training labels for multi-object editing.

With this source of training data, we propose Neural Assets – per object latent representations with consistent 3D appearance but variable 3D pose. Neural Assets are trained by extracting their visual appearances from one frame in a video and reconstructing their appearances in a different frame in the video conditioned on the corresponding 3D bounding boxes. This supports learning consistent 3D appearance disentangled from 3D pose. We can then tokenize any number of Neural Assets and feed this sequence to a fine-tuned conditional image generator for precise, multi-object, 3D control.

**Our main contributions** are threefold: **(i)** A Neural Asset formulation that represent objects with disentangled appearance and pose features. By training on paired video frames, it enables fine-grained 3D control of individual objects. **(ii)** Our framework is applicable to both synthetic and real-world scenes, achieving state-of-the-art results on 3D-aware object editing. **(iii)** We extend Neural Assets to further support compositional scene generation, such as swapping the background of two scenes and transferring objects across scenes. We show the versatile control ability of our model in Fig. 1.

## 2   Related Work

**2D spatial control in diffusion models (DMs).** With the rapid growth of diffusion-based visual generation [22, 44, 70, 79, 82, 94], there have been many works aiming to inject spatial control to pre-trained DMs via 2D bounding boxes or segmentation masks. One line of research achieves this by manipulating text prompts [11, 32, 51], intermediate attention maps [12, 16, 17, 27, 41, 52, 110] or noisy latents [25, 64, 69, 90] in the diffusion process, without the need to change model weights. Closer to ours are methods that fine-tune pre-trained DMs to support additional spatial conditioning inputs [5, 29, 33, 45, 112, 114]. GLIGEN [59] introduces new attention layers to condition on bounding boxes. InstanceDiffusion [103] further supports object masks, points, and scribbles with a unified feature fusion block. To incorporate dense control signals such as depth maps and surface normals, ControlNet [116] adds zero-initialized convolution layers around the original network blocks. Recently, Boximator [99] demonstrates that such 2D control can be extended to video models with a similar technique. Several existing works [3, 107] leverage natural motion observed in video and similar to our work propose to train on paired video frames to achieve pixel-level control. In our work, we build upon pre-trained DMs and leverage 3D bounding boxes as spatial conditioning, which enables 3D-aware control such as object rotation and occlusion handling.

**3D-aware image generation.** Earlier works leverage differentiable rendering to learn 3D Generative Adversarial Networks (GANs) [34] from monocular images, with explicit 3D representations such as radiance fields [14, 15, 37, 71, 86, 111] and meshes [18, 19, 31, 73, 74]. Inspired by the great success of DMs in image generation, several works try to lift the 2D knowledge to 3D [49, 60, 62, 66, 75, 89, 98, 105]. The pioneering work 3DiM [105] and follow-up work Zero-1-to-3 [61] directly train diffusion models on multi-view renderings of 3D assets. However, this line of research only considers single objects without background, which cannot handle in-the-wild data with complex backgrounds. Closest to ours are methods that process multi-object real-world scenes [2, 72, 84, 115]. OBJect-3DIT [67] studies language-guided 3D-aware object editing by training on paired synthetic data, limiting its performance on real-world images [115]. LooseControl [8] converts 3D bounding boxes to depth maps to guide the object pose. Yet, it cannot be directly applied to edit existing images. In contrast, our Neural Asset representation captures both object appearance and 3D pose. It can be easily trained on real-world videos to achieve multi-object 3D edits.

From a methodology perspective, there have been prior works learning disentangled appearance and pose representations for 3D-aware multi-object image editing [71, 100, 111]. However, they are all based on the GAN framework [34] and do not learn generalizable object representations via an encoder. In contrast, we build upon a large-scale pre-trained image diffusion model [79] and powerful feature extractors [13], enabling editing of complex real-world scenes.

**Personalized image generation.** Since the seminal works DreamBooth [81] and Textual Inversion [30] which perform personalized generation via test-time fine-tuning, huge efforts have been made to achieve this in a zero-shot manner [47, 58, 88, 101, 106, 109]. Most of them are only able to synthesize one subject, and cannot control the spatial location of the generated instance. A notable exception is Subject-Diffusion [65], which leverages frozen CLIP embeddings for object appearance and 2D bounding boxes for object position. Still, it cannot explicitly control the 3D pose of objects.

**Object-centric representation learning.** Our Neural Asset representation is also related to recent object-centric slot representations [48, 63, 91, 92, 108] that decompose scenes into a set of object entities. Object slots provide a useful interface for editing such as object attributes [93], motions [87], 3D poses [46], and global camera poses [83]. Nevertheless, these models show significantly degraded results on real-world data. Neural Assets also consist of disentangled appearance and pose features of objects. Different from existing slot-based models, we fine-tune self-supervised visual encoders and connect them with large-scale pre-trained DMs, which scales up to complex real-world data.

## 3 Method: Neural Assets

Inspired by 3D assets in computer graphics software, we propose Neural Assets as learnable object-centric representations. A Neural Asset comprises an appearance and an object pose representation, which is trained to reconstruct the object via conditioning a diffusion model (Sec. 3.2). Trained on paired images, our method learns disentangled representations, enabling 3D-aware object editing and compositional generation at inference time (Sec. 3.3). Our framework is summarized in Fig. 2.

### 3.1 Background: 3D Assets in Computer Graphics

3D object models, or *3D assets*, are basic components of any 3D scene in computer graphics software, such as Blender [20]. A typical workflow includes selecting $N$ 3D assets $\{\hat{a}_1, ..., \hat{a}_N\}$ from an asset library and placing them into a scene. Formally, one can define a 3D asset as a tuple $\hat{a}_i \triangleq (\mathcal{A}_i, \mathcal{P}_i)$, where $\mathcal{A}_i$ is a set of descriptors defining the asset's appearance (e.g., canonical 3D shape and surface textures) and $\mathcal{P}_i$ describes its pose (e.g., rigid transformation and scaling from its canonical pose).

### 3.2 Neural Assets

Inspired by 3D assets in computer graphics, our goal is to enable such capabilities (i.e., 3D control and compositional generation) in recent generative models. To achieve this, we define a *Neural Asset* as a tuple $a_i \triangleq (A_i, P_i)$, where $A_i \in \mathbb{R}^{(K \times D)}$ is a flattened sequence of $K$ $D$-dimensional vectors describing the appearance of an asset, and $P_i \in \mathbb{R}^{D'}$ is a $D'$-dimensional embedding of the asset's pose in a scene. In other words, a Neural Asset is fully described by learnable embedding vectors, factorized into appearance and pose. This factorization enables independent control over appearance and pose of an asset, similar to how 3D object models can be controlled in traditional computer

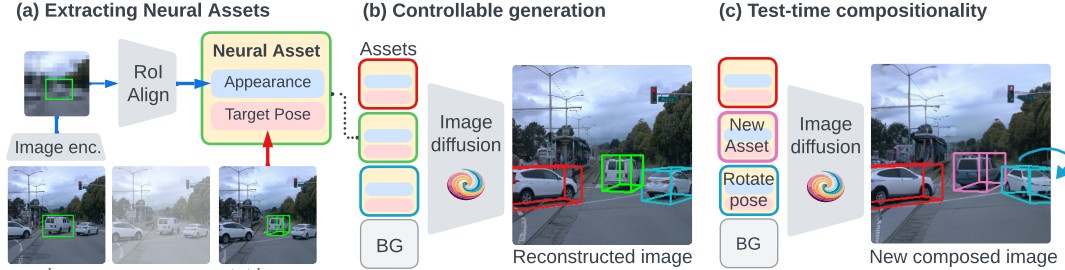

**(a) Extracting Neural Assets**    **(b) Controllable generation**    **(c) Test-time compositionality**

Figure 2: **Neural Assets framework**. (a) We train our model on pairs of video frames, which contain objects under different poses. We encode appearance tokens from a source image with RoIAlign, and pose tokens from the objects' 3D bounding boxes in a target image. They are combined to form our Neural Asset representations. (b) An image diffusion model is conditioned on Neural Assets and a separate background token to reconstruct the target image as the training signal. (c) During inference, we can manipulate the Neural Assets to control the objects in the generated image: rotate the object's pose (blue) or replace an object by a different one from another image (pink).

graphics software. Importantly, besides the 3D pose of assets, our approach does not require any explicit mapping of objects into 3D, such as depth maps or the NeRF representation [68].

### 3.2.1 Asset Encoding

In the following, we describe how both the appearance $A_i$ and the pose $P_i$ of a Neural Asset $a_i$ are obtained from visual observations (such as an image or a frame in a video). Importantly, the appearance and pose representations are not necessarily encoded from the same observation, i.e., they can be encoded from two separate frames sampled from a video. We find this strategy critical to learn disentangled and controllable representations, which we will discuss in detail in Sec. 3.3.

**Appearance encoding.** At a high level, we wish to obtain a set of $N$ Neural Asset appearance tokens $A_i$ from a visual observation $x_{\text{src}}$, where $x_{\text{src}}$ can be an image or a frame in a video. While one could approach this problem in a fully-unsupervised fashion, using a method such as Slot Attention [63] to decompose an image into a set of object representations, we choose to use readily-available annotations to allow fine-grained specification of objects of interest. In particular, we assume that a 2D bounding box $b_i$ is provided for each Neural Asset $a_i$, specifying which object should be extracted from $x_{\text{src}}$. Therefore, we obtain the appearance representation $A_i$ as follows:

$$A_i = \text{Flatten}(\text{RoIAlign}(H_i, b_i)), \quad H_i = \text{Enc}(x_{\text{src}}), \quad (1)$$

where $H_i$ is the output feature map of a visual encoder $\text{Enc}$. RoIAlign [38] extracts a fixed size feature map using the provided bounding box $b_i$ which is flattened to form the appearance token $A_i$. This factorization allows us to extract $N$ object appearances from an image with just one encoder forward pass. In contrast, previous methods [65, 109] crop each object out to extract features separately, and thus requires $N$ encoder passes. This becomes unaffordable if we jointly fine-tune the visual encoder, which is key to learning generalizable features as we will show in the ablation study.

**Pose encoding.** The pose token $P_i$ of a Neural Asset $a_i$ is the primary interface for controlling the presence and 3D pose of an object in the rendered scene. In this work, we assume that the object pose is provided in terms of a 3D bounding box, which fully specifies its location, orientation, and size in the scene. Formally, we take four corners spanning the 3D bounding box[1] and project them to the image plane to get $\{c_i^j = (h_i^j, w_i^j, d_i^j)\}_{j=1}^4$, with the projected 2D coordinate $(h_i^j, w_i^j)$, and the 3D depth $d_{i,j}$. We obtain the pose representation $P_i$ for a Neural Asset as follows:

$$P_i = \text{MLP}(C_i), \quad C_i = \text{Concat}[c_i^1, c_i^2, c_i^3, c_i^4], \quad (2)$$

where we first concatenate the four corners $c_i^j$ to form $C_i \in \mathbb{R}^{12}$, and then project it to $P_i \in \mathbb{R}^{D'}$ via an MLP. We tried the Fourier coordinate encoding in prior works [59, 103] but did not find it helpful.

There are alternative ways to represent 3D bounding boxes (e.g., concatenation of center, size, and rotation commonly used in 3D object detection [57]), which we compare in Appendix B.4. In this

---

[1]Only three corners are needed to fully define a 3D bounding box, but we found a 4-corner representation beneficial to work with. Previous research [118] also shows that over-parametrization can benefit model learning.

work, we assume the availability of training data with 3D annotations – obtaining high-quality 3D object boxes for videos at scale is still an open research problem, but may soon be within reach given recent progress in monocular 3D detection [102], depth estimation [7, 113], and pose tracking [9].

**Serialization of multiple Neural Assets.** We encode a set of $N$ Neural Assets into a sequence of tokens that can be appended to or used in place of text embeddings for conditioning a generative model. In particular, we first concatenate the appearance token $A_i$ and the pose token $P_i$ channel-wise, and then linearly project it to obtain a Neural Asset representation $a_i$ as follows:

$$a_i = \mathrm{Linear}(\tilde{a}_i), \quad \tilde{a}_i = \mathrm{Concat}[A_i, P_i] \in \mathbb{R}^{K \times D + D'}. \tag{3}$$

Channel-wise concatenation uniquely binds one pose token with one appearance representation in the presence of multiple Neural Assets. An alternative solution is to learn such association with positional encoding. Yet, it breaks the permutation-invariance of the generator against the order of input objects and leads to poor results in our preliminary experiments. Finally, we simply concatenate multiple Neural Assets along the token axis to arrive at our token sequence, which can be used as a drop-in replacement for a sequence of text tokens in a text-to-image generation model.

**Background modeling.** Similar to prior works [71, 111], we found it helpful to encode the scene background separately, which enables independent control thereof (e.g., swapping out the scene, or controlling global properties such as lighting). We choose the following heuristic strategy to encode the background: to avoid leakage of foreground object information, we mask all pixels within asset bounding boxes $b_i$. We then pass this masked image through the image encoder $\mathrm{Enc}$ (shared weights with the foreground asset encoder) and apply a global $\mathrm{RoIAlign}$, i.e., using the entire image as region of interest, to obtain a background appearance token $A_{\mathrm{bg}} \in \mathbb{R}^{(K \times D)}$. Similar to a Neural Asset, we also attach a pose token $P_{\mathrm{bg}}$ to $A_{\mathrm{bg}}$. This can either be a timestep embedding of the video frame (relative to the source frame) or a relative camera pose embedding, if available. In the serialized representations, the background token is treated the same as Neural Assets, i.e., we concatenate $A_{\mathrm{bg}}$ and $P_{\mathrm{bg}}$ channel-wise and linearly project it. Finally, the foreground assets $a_i$ and the background token are concatenated along the token dimension and used to condition the generator.

### 3.2.2 Generative Decoder

To generate images from Neural Assets, we make minimal assumptions about the architecture or training setup of the generative image model to ensure compatibility with future large-scale pre-trained image generators. In particular, we assume that the generative image model accepts a sequence of tokens as conditioning signal: for most base models this would be a sequence of tokens derived from text prompts, which we can easily replace with a sequence of Neural Asset tokens.

As a representative for this class of models, we adopt Stable Diffusion v2.1 [79] for the generative decoder. See Appendix C for details on this model. Starting from the pre-trained text-to-image checkpoint, we fine-tune the entire model end-to-end to accept Neural Assets tokens instead of text tokens as conditioning signal. The training and inference setup is explained in the following section.

### 3.3 Learning and Inference

**Learning from frame pairs.** As outlined in the introduction, we require a scalable data source of object-level "edits" in 3D space to effectively learn multi-object 3D control capabilities. Video data offers a natural solution to this problem: as the camera and the content of the scene moves or changes over time, objects are observed from various view points and thus in various poses and lighting conditions [78]. We exploit this signal by randomly sampling pairs of frames from video clips, where we take one frame as the "source" image $x_{\mathrm{src}}$ and the other frame as the "target" image $x_{\mathrm{tgt}}$.

As described earlier, we obtain the appearance token $A_i$ of Neural Assets from the *source* frame $x_{\mathrm{src}}$ by extracting object features using 2D box annotations. Next, we obtain the pose token $P_i$ for each extracted asset from the *target* frame $x_{\mathrm{tgt}}$, for which we need to identify the correspondences between objects in both frames. In practice, such correspondences can be obtained, for example, by applying an object tracking model on the underlying video. Finally, with the associated appearance and pose representations, we condition the image generator on them and train it to reconstruct the target frame $x_{\mathrm{tgt}}$, i.e., using the denoising loss of Stable Diffusion v2.1 in our case. Such a paired frame training strategy forces the model to learn an appearance token that is invariant to object pose and leverage the pose token to synthesize the new object, avoiding the trivial solution of simple pixel-copying.

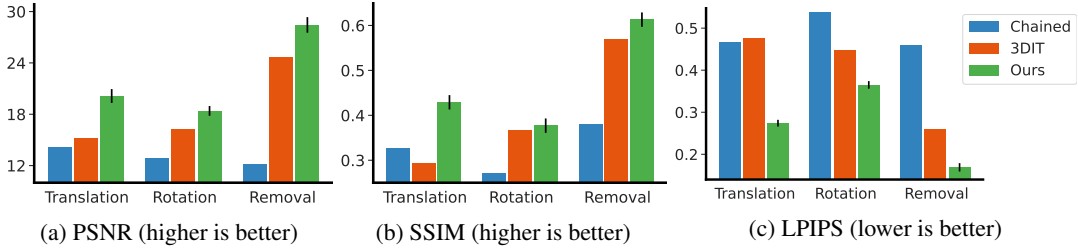

| (a) PSNR (higher is better) | (b) SSIM (higher is better) | (c) LPIPS (lower is better) |

Figure 3: **Single-object editing results on OBJect unseen object subset**. We evaluate on the *Translation*, *Rotation*, and *Removal* tasks. We follow 3DIT [67] to compute metrics inside the edited object's bounding box. Our results are averaged over 3 random seeds.

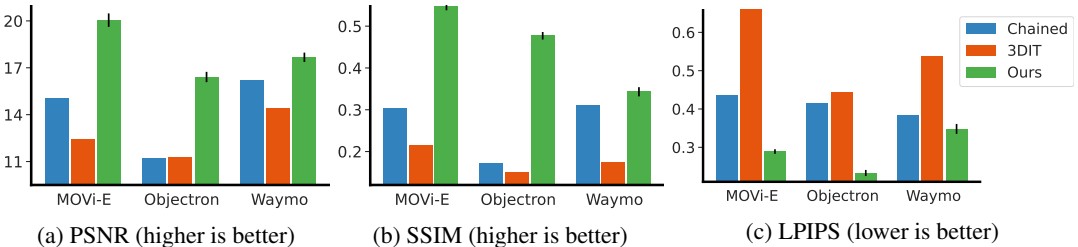

| (a) PSNR (higher is better) | (b) SSIM (higher is better) | (c) LPIPS (lower is better) |

Figure 4: **Multi-object editing results on MOVi-E, Objectron, and Waymo Open** (denoted as Waymo in the figures). We compute metrics inside the edited objects' bounding boxes.

**Test-time controllability.** The learned disentangled representations naturally enable multi-object scene-level editing as we will show in Sec. 4.3. Since we encode 3D bounding boxes to pose tokens $P_i$, we can move, rotate, and rescale objects by changing the box coordinates. We can also compose Neural Assets $a_i$ across scenes to generate new scenes. In addition, our background modeling design supports swapping the environment map of the scene. Importantly, as we will see in the experiments, our image generator learns to naturally blend the objects into their new environment at new positions, with realistic lighting effects such as rendering and adapting shadows correctly.

## 4 Experiments

In this section, we conduct extensive experiments to answer the following questions: **(i)** Can Neural Assets enable accurate 3D object editing? **(ii)** What practical applications does our method support on real-world scenes? **(iii)** What is the impact of each design choice in our framework?

### 4.1 Experimental Setup

**Datasets.** We select four datasets with object or camera motion, which span different levels of complexity. *OBJect* [67] is introduced in 3DIT [67], which is one of our baselines. It contains 400k synthetic scenes rendered by Blender [20] with a static camera. Up to four Objaverse [21] assets are placed on a textured ground and only one object is randomly moved on the ground. For a fair comparison with 3DIT, we use 2D bounding boxes plus rotation angles as object poses, and follow them to base our model on Stable Diffusion v1.5 [79]. *MOVi-E* [36] consists of Blender simulated videos with up to 23 objects. It is more challenging than OBJect as it has linear camera motion and there can be multiple objects moving simultaneously. *Objectron* [1] is a big step up in complexity as it captures real-world objects with complex backgrounds. 15k object-centric videos covering objects from nine categories are recorded with $360°$ camera movement. *Waymo Open* [97] is a real-world self-driving dataset captured by car mounted cameras. We follow prior work [111] to use only the front view and filter out cars that are too small. See Appendix A.1 for more details on datasets.

**Baselines.** We compare to methods that can perform 3D-aware editing on existing images and have released their code. *3DIT* [67] fine-tunes Zero-1-to-3 [61] on the OBJect dataset to support translation and rotation of objects. However, it cannot render big viewpoint changes as it does not encode camera poses. Following [67], we create another baseline (dubbed *Chained*) by using SAM [55] to segment the object of interest, removing it using Stable Diffusion inpainting model [79], running Zero-1-to-3

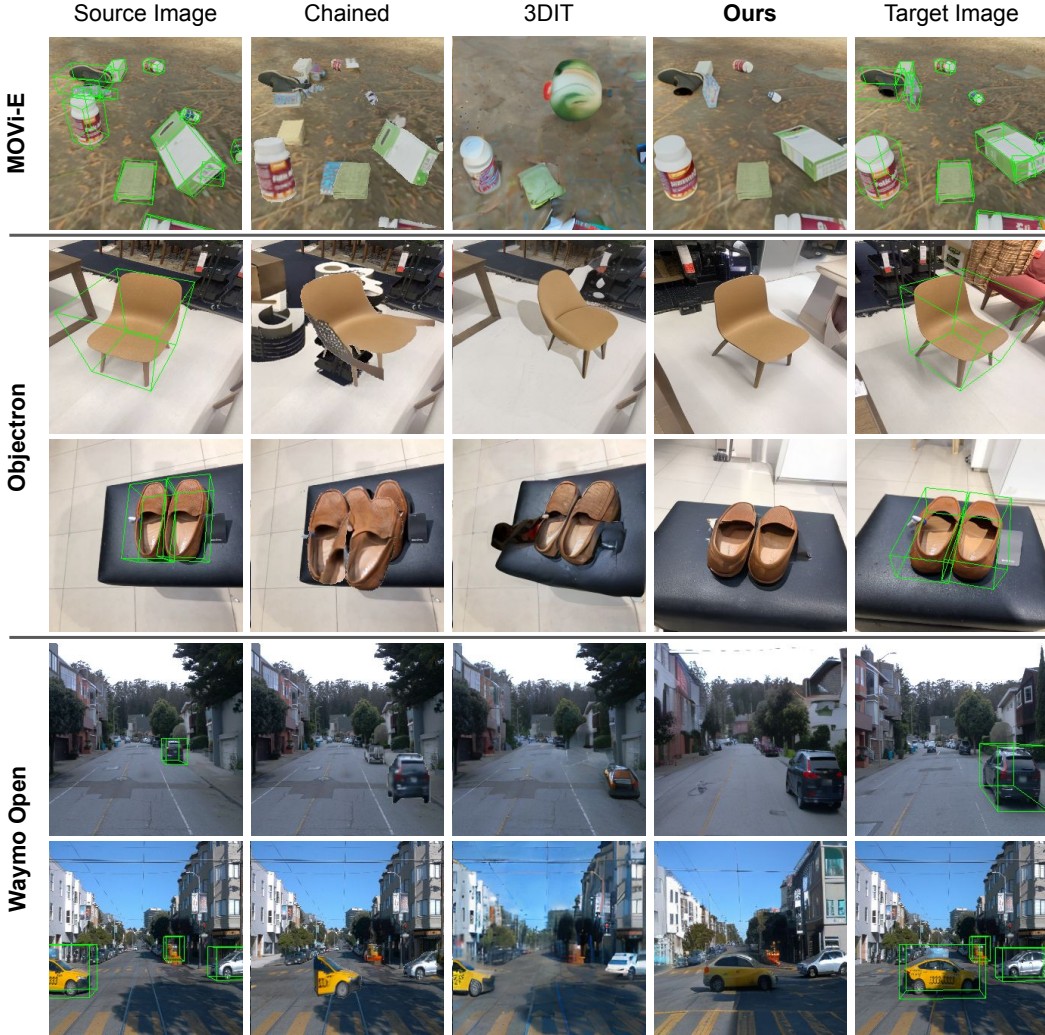

Figure 5: **Qualitative comparison on MOVi-E, Objectron, and Waymo Open**. All models generate a new image given a source image and the 3D bounding box of target objects. Our method performs the best in object identity preservation, editing accuracy, and background modeling.

to rotate and scale the object, and stitching it to the target position. Since none of these baselines can control multiple objects simultaneously, we apply them to edit all objects sequentially.

**Evaluation settings.** We report common metrics to measure the quality of the edited image – PSNR, SSIM [104], LPIPS [117], and FID [42]. Following prior works [49, 67], we also compute object-level metrics on cropped out image patches of edited objects. To evaluate the fidelity of edited objects, we take the DINO [13] feature similarity metric proposed in [81]. On video datasets, we randomly sample source and target images in each testing video and fix them across runs for consistent results.

**Implementation Details.** For all experiments, we resize images to $256 \times 256$. DINO self-supervised pre-trained ViT-B/8 [13] is adopted as the visual encoder Enc, and jointly fine-tuned with the generator. All our models are trained using the Adam optimizer [53] with a batch size of $1536$ on $256$ TPUv4 chips. For inference, we generate images by running the DDIM [95] sampler for 50 steps. For more training and inference details, please refer to Appendix A.4.

### 4.2 Main Results

**Single-object editing.** We first compare the ability to control the 3D pose of a single object on the OBJect dataset. Fig. 3 presents the results on the unseen object subset. We do not show FID here as it mainly measures the visual quality of generated examples, which does not reflect the editing accuracy.

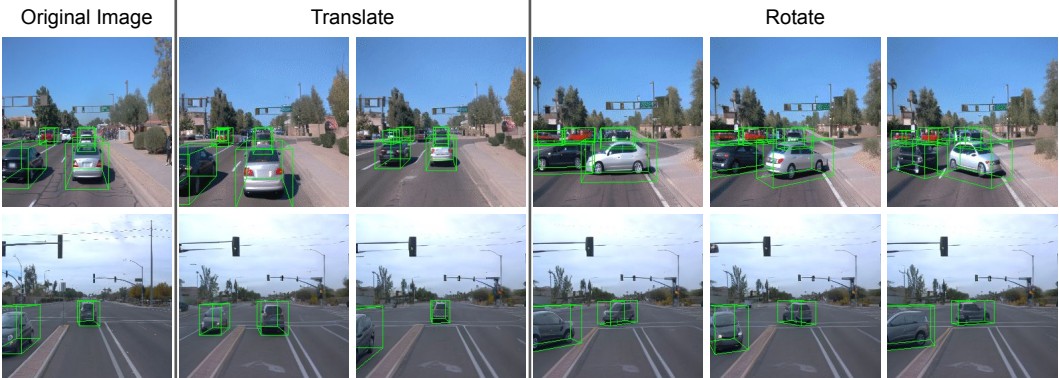

Figure 6: **Object translation and rotation** by manipulating 3D bounding boxes on Waymo Open. See our project page for videos and additional object rescaling results.

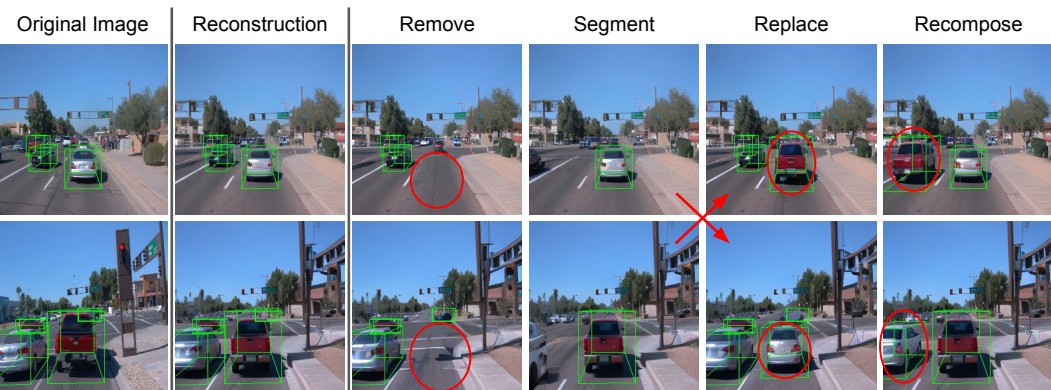

Figure 7: **Compositional generation results** on Waymo Open. By composing Neural Assets, we can remove and segment objects, as well as transfer and recompose objects between scenes.

For results on the seen object subset and FID, please refer to Appendix B.1, where we observe similar trends. Compared to baselines, our model does not condition on text (e.g., the category name of the object to edit) as in 3DIT and is not trained on curated multi-view images of 3D assets as in Zero-1-to-3. Still, we achieve state-of-the-art performance on all three tasks. This is because our Neural Assets representation learns disentangled appearance and pose features, which is able to preserve object identity while changing its placement smoothly. Also, the fine-tuned DINO encoder generalizes better to unseen objects compared to the frozen CLIP visual encoder used by baselines.

**Multi-object editing.** Fig. 4 shows the results on MOVi-E, Objectron, and Waymo Open, where multiple objects are manipulated in each sample. Similar to the single-object case, we compute metrics inside the object bounding boxes, and leave the image-level results to Appendix B.1. Our model outperforms baselines by a sizeable margin across datasets. Fig. 5 presents the qualitative results. When there are multiple objects of the same class in the scene (e.g., boxes in the MOVi-E example and cars on Waymo Open), 3DIT is unable to edit the correct instance. In addition, it generalizes poorly to real-world scenes. Thanks to the object cropping step, Chained baseline can identify the correct object of interest. However, the edited object is simply pasted to the target location, leading to unrealistic appearance due to missing lighting effects such as shadows. In contrast, our model is able to control all objects precisely, preserve their fidelity, and blend them into the background naturally. Since we encode the camera pose, we can also model global viewpoint change as shown in the third row. See Appendix B.1 for additional qualitative results.

### 4.3 Controllable Scene Generation

In this section, we show versatile control of scene objects on Waymo Open. For results on Objectron, please refer to Appendix B.3. As shown in Fig. 6, we can translate and rotate cars in driving scenes. The model understands the 3D world as objects zoom in and out when moving, and show consistent

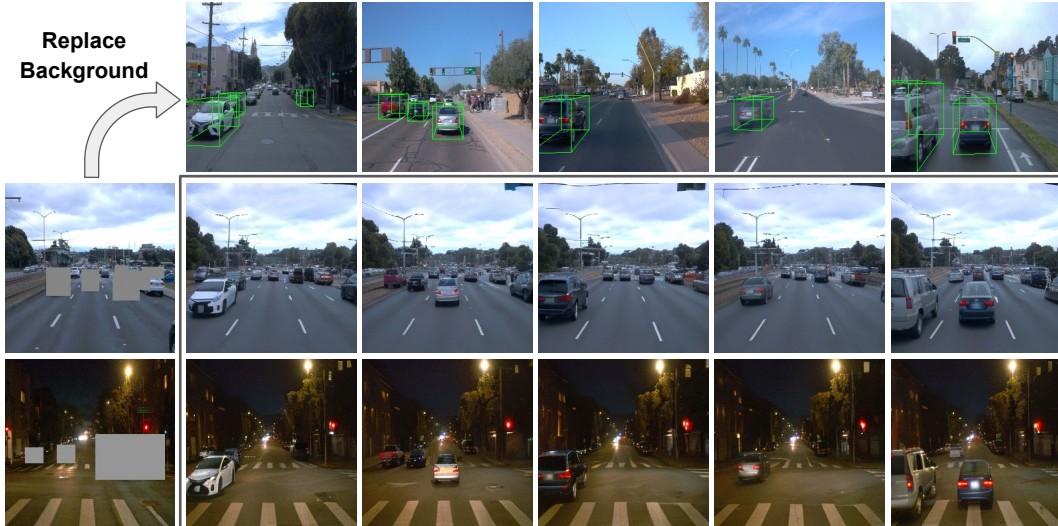

**Replace Background**

Figure 8: **Transfer backgrounds between scenes** by replacing the background token on Waymo Open. The objects can adapt to new environments, e.g., the car lights are turned on at night.

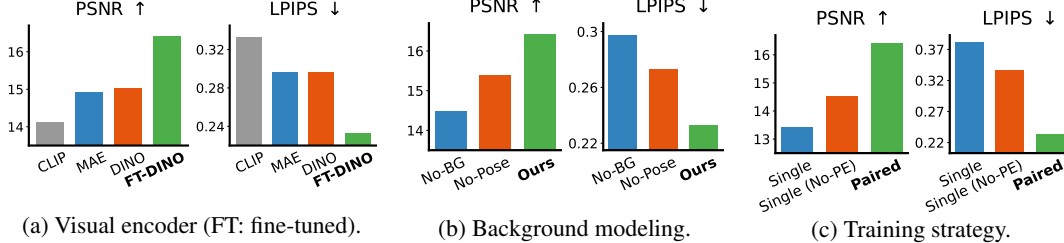

(a) Visual encoder (FT: fine-tuned).  (b) Background modeling.  (c) Training strategy.

Figure 9: **Comparison of (a) visual encoders, (b) background modeling, and (c) training strategies on Objectron**. Bold entry denotes our full model. See text for each variant. We report PSNR and LPIPS computed within object bounding boxes, and leave other metrics to Appendix B.2.

novel views when rotating. Fig. 7 presents our ability of compositional generation, where objects are removed, segmented out, and transferred across scenes. Notice how the model handles occlusion and inpaints the scene properly. Finally, Fig. 8 demonstrates background swapping between scenes. The generator is able to harmonize objects with the new environment. For example, the car lights are turned on and rendered with specular highlight when using a background image from a night scene.

## 4.4  Ablation Study

We study the effect of each component in the model. All ablations are run on Objectron since it is a real-world dataset with complex background, and has higher object diversity than Waymo Open.

**Visual encoder.** Previous image-conditioned diffusion models [49, 61, 62] usually use the frozen image encoder of CLIP [76] to extract visual features. Instead, as shown in Fig. 9a, we found that both MAE [39] and DINO [13] pre-trained ViTs give better results. This is because CLIP's image encoder only captures high-level semantics of images, which suffices in single-object tasks, but fails in our multi-object setting. In contrast, MAE and DINO pre-training enable the model to extract more fine-grained features. Besides, DINO outperforms MAE as its features contain richer 3D information, which aligns with recent research [6]. Finally, jointly fine-tuning the image encoder learns more generalizable appearance tokens in Neural Assets, leading to the best performance.

**Background modeling.** We compare our full model with two variants: **(i)** not conditioning on any background tokens (dubbed *No-BG*), and **(ii)** conditioning on background appearance tokens but not using relative camera pose as pose tokens (dubbed *No-Pose*). As shown in Fig. 9b, our background modeling strategy performs the best in image-level metrics as backgrounds usually occupy a large part of real-world images. Interestingly, our method also achieves significantly better object-level metrics. This is because given background appearance and pose, the model does not need to infer them from object tokens, leading to more disentangled Neural Assets representations.

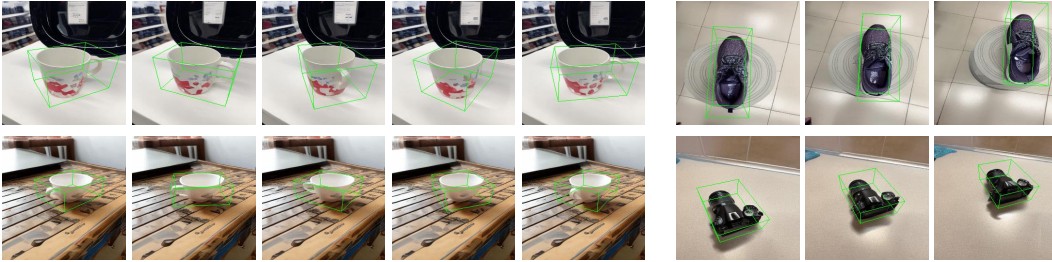

|                                              |                                          |
|:--------------------------------------------:|:----------------------------------------:|
| (a) Symmetry ambiguity when rotating objects | (b) Object-camera motion entanglement    |

Figure 10: **Failure case analysis**. Our model mainly has two failure cases: (a) symmetry ambiguity, where the handle of the cup gets flipped when it rotates by 180 degrees; (b) camera-object motion entanglement, where the background also moves when we translate the foreground object. Both issues will likely be resolved if we train our Neural Assets model on more diverse data.

**Training strategy.** As described in Sec. 3.3, we train on videos and extract appearance and pose tokens from different frames. We compare such design with training on a single frame in Fig. 9c. Our paired frame training strategy clearly outperforms single frame training. Since the appearance token is extracted by a ViT with positional encoding, it already contains object position information, which acts as a shortcut for image reconstruction. Therefore, the model ignores the input object pose token, resulting in poor controllability. One way to alleviate this is removing the positional encoding in the image encoder (dubbed *NO-PE*), which still underperforms paired frame training. This is because to reconstruct objects with visual features extracted from a different frame, the model is forced to infer their underlying 3D structure instead of simply copying pixels. In addition, the generator needs to render realistic lighting effects such as shadows under the new scene configuration.

## 5    Conclusion

In this paper, we present Neural Assets, vector-based representations of objects and scene elements with disentangled appearance and pose features. By connecting with pre-trained image generators, we enable controllable 3D scene generation. Our method is capable of controlling multiple objects in the 3D space as well as transferring assets across scenes, both on synthetic and real-world datasets. We view our work as an important step towards general-purpose neural-based simulators.

**Limitations and future works.** One main failure case of our model is symmetry ambiguity. As can be seen from the rotation results in Fig. 10 (a), the handle of the cup gets flipped when it rotates by 180 degree. Another failure case that only happens on Objectron is the entanglement of global camera motion and local object movement (Fig. 10 (b)). This is because Objectron videos only contain camera motion while objects always stay static. Both issues will likely be resolved if we train our model on larger-scale datasets with more diverse object and camera motion.

An ideal Neural Asset should enable control over all potential configurations of an object such as deformation (e.g., a walking cat), rigid articulation (e.g., opening of a scissor), and structural decomposition (e.g., tomatoes being cut). In this work, we first tackle the foremost important aspect, i.e., controlling 3D rigid object pose and background composition which applies to almost all the objects. Hence our current method does not allow for controlling structural changes. However, it can be adapted when suitable datasets are developed that capture other changes in objects.

Another limitation is that our approach is currently limited to existing datasets that have 3D bounding box annotations. Yet, with recent advances in vision foundation models [9, 55, 113], we may soon have scalable 3D annotation pipelines similar to their 2D counterparts. One notable example is OmniNOCS [56], which works on both Waymo and Objectron (datasets we used in this work), and *diverse, in-the-wild Internet images* for a wide range of object classes. It can be used to create larger open domain datasets to learn Neural Assets. We see this as an interesting future direction.

## Acknowledgements

We would like to thank Etienne Pot, Klaus Greff, Shlomi Fruchter, and Amir Hertz for their advise regarding infrastructure. We would further like to thank Mehdi S. M. Sajjadi, João Carreira, Sean Kirmani, Yi Yang, Daniel Zoran, David Fleet, Kevin Murphy, and Mike Mozer for helpful discussions.

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

# A    Detailed Experimental Setup

In this section, we provide full details on the datasets, baselines, evaluation settings, and the training and inference implementation of our model.

## A.1    Datasets

**OBJect [67]** consists of Blender [20] rendered scenes where multiple (up to four) objects are placed on a flat textured ground. The objects come from a 59k subset of Objaverse dataset [21]. A total of 18 background maps are used to provide environmental lighting. Four types of object-level editing are provided – translation, rotation, removal, and insertion, each with 100k simulated data. Notably, only one object is edited in each data, and the translation and rotation is always on the ground (i.e., perpendicular to the gravity vector). For a fair comparison with the 3DIT baseline [67], we use the 2D rotated bounding box to represent object pose, which is composed of two corners of the 2D bounding box and the rotation angle over the gravity axis. This dataset is under the Open Data Commons Attribution License (ODC-By)[2].

**MOVi-E [36]** contains 10k videos simulated using Kubric [36]. Each scene contains 11 to 23 real-world objects from the Google Scanned Objects (GSO) repository [23]. At the start of each video, several objects are thrown to the ground to collide with other objects. Similar to OBJect, environmental lighting is provided by a randomly sampled environment map image. The camera follows a small linear motion. The full data generation pipeline is under the Apache 2.0 license[3].

**Objectron [1]** contains 15k object-centric video clips of common daily objects covering nine categories. Each video comes with object pose tracking throughout the video, and we process it to obtain 3D bounding boxes. Since this dataset does not provide 2D bounding box labels, we project the eight corners of 3D boxes to the image, and take the tight bounding box of projected points as 2D boxes. Objectron is licensed under the Computational Use of Data Agreement 1.0 (C-UDA-1.0)[4].

**Waymo Open [97].** The Waymo Open Dataset consists of 1k videos of self-driving scenes recorded by car mounted cameras. Following prior works [24, 111], we take the front view camera and bounding box annotations of cars. Notably, the 3D bounding boxes only have a heading angle (rotation along the yaw-axis) annotation, and thus we treat the other two rotation angles as 0. Besides, the provided 2D boxes and 3D boxes are not aligned, preventing us from doing paired frame training. We instead project 3D boxes to get associated 2D boxes similar to on Objectron. Waymo Open is licensed under the Waymo Dataset License Agreement for Non-Commercial Use (August 2019)[5].

**Data Pre-processing.** For all datasets, we resize the images to $256 \times 256$ regardless of the original aspect ratio. On Objectron, we discard all videos from the bike class as it contains many blurry frames and inaccurate 3D bounding box annotations. On Waymo, we remove all cars whose 2D bounding box is smaller than 1% of the image area. We do not apply data augmentation except on Waymo Open, where we apply random horizontal flip and random resize crop following [24].

## A.2    Baselines

**3DIT [67]** fine-tunes Zero-1-to-3 [61] to support scene-level 3D object edits. We generate the editing instruction from the target object pose, such as the translation coordinate and the rotation angle. However, this method does not support large viewpoint changes as it does not encode camera poses. We take their official code and pre-trained weights of the Multitask variant. 3DIT is under the CreativeML Open RAIL-M license[6].

**Chained.** This baseline is inspired by [67, 72], where we chain multiple models together to achieve 3D-aware object editing. An editing step usually contains three steps: **(i)** crop out the object of interest and inpaint its region with backgrounds, **(ii)** synthesize the object under the new pose, and **(iii)** place the object to the new location. For **(i)**, we apply SAM [55] to segment the object using 2D bounding box prompt, and inpaint the original object region with Stable Diffusion v2 inpainting

---

[2]https://huggingface.co/datasets/allenai/object-edit/blob/main/README.md
[3]https://github.com/google-research/kubric/blob/main/LICENSE
[4]https://github.com/google-research-datasets/Objectron#license
[5]https://waymo.com/open/terms
[6]https://github.com/allenai/object-edit/blob/main/LICENSE

model [79]. For **(ii)**, we run Zero-1-to-3 [61] to re-pose the object according to the target 3D bounding box. For **(iii)**, following [61], we first get the alpha mask of the re-posed object using an online tool[7], and insert it to the new position via alpha blending. It is worth noting that Zero-1-to-3 does not support camera rotation over the roll axis. For all models, we take their official code and pre-trained weights. SAM is under the Apache 2.0 license[8]. Stable Diffusion v2 inpainting model is under the CreativeML Open RAIL++-M License[9]. Zero-1-to-3 is under the MIT license[10]. The online alpha mask extraction tool is under the Apache 2.0 license[11].

### A.3 Evaluation Settings

We report PSNR, SSIM [104], LPIPS [117], and FID [42] to measure the accuracy of the edited image. We compute metrics both on the entire image, and within the 2D bounding box of edited objects. For box-level metrics, we follow [67] to crop out each object and directly run the metric without resizing. We also evaluate the identity preservation of objects using the DINO [13] feature similarity proposed in [81], which runs a DINO self-supervised pre-trained ViT on cropped object patches to extract features and compute the cosine similarity between predicted and ground-truth image.

### A.4 Our Implementation Details

**Model architecture.** We take Stable Diffusion (SD) v2.1 [79] as our image generator except for experiments on the OBJect dataset, where we use SD v1.5 for a fair comparison with baselines. Similar to prior works [49, 89], we also observe clearly better performance using SD v2.1 compared to v1.5. However, we note that our Neural Assets framework generalizes to any image generator that conditions on a sequence of tokens. We implement the visual encoder $\mathrm{Enc}$ with a DINO self-supervised pre-trained ViT-B/8 [13], which outputs a feature map of shape $28 \times 28$ given a $256 \times 256$ image. For each object, we apply RoIAlign [38] to extract a $2 \times 2$ small feature map and flatten it, i.e., the appearance token $A_i$ has a sequence length of $K = 4$. Since the conditioning token dimension of pre-trained SD v2.1 is 1024, we use a two-layer MLP to transform the 3D bounding boxes input to $D' = 1024$, and linearly project the concatenated appearance and pose token back to 1024. For background modeling, we mask all pixels within object boxes by setting them to a fixed value of 0.5, and extract features with the same DINO encoder. Instead, the pose token is obtained by applying a different two-layer MLP on the relative camera pose between the source and the target image.

**Training.** We implement the entire Neural Assets framework in JAX [10] using the Flax [40] neural network library. We train all model components jointly using the Adam optimizer [53] with a batch size of 1536 on 256 TPUv5 chips (16GB memory each). We use a peak learning rate of $5 \times 10^{-5}$ for the image generator and the visual encoder, and a larger learning rate of $1 \times 10^{-3}$ for remaining layers (MLPs and linear projection layers). Both learning rates are linearly warmed up in the first 1,000 steps and stay constant. A gradient clipping of 1.0 is applied to stabilize training. We found that the model overfits more severely on real-world data with complex backgrounds compared to synthetic datasets. Therefore, we train the model for 200k steps on OBJect and MOVi-E which takes 24 hours, and 50k steps on Objectron and Waymo Open which takes 6 hours. In order to apply classifier-free guidance (CFG) [43], we randomly drop the appearance and pose token (i.e., setting them as zeros) with a probability of 10%. CFG improves the performance and also alleviates overfitting in training.

**Inference.** We run the DDIM sampler [95] for 50 steps to generate images. We found the model works well with CFG scale between 1.5 and 4, and thus choose to use 2.0 in all the experiments.

---

[7] https://github.com/OPHoperHPO/image-background-remove-tool
[8] https://github.com/facebookresearch/segment-anything#license
[9] https://huggingface.co/stabilityai/stable-diffusion-2-inpainting
[10] https://github.com/cvlab-columbia/zero123/blob/main/LICENSE
[11] https://github.com/OPHoperHPO/image-background-remove-tool/blob/master/LICENSE

Table 1: **Single-object editing results on OBJect**. We evaluate on the *Translation*, *Rotation*, and *Removal* tasks. We follow 3DIT [67] to evaluate on both seen and unseen object subsets, and compute metrics inside the edited object's bounding box. Our results are averaged over 3 random seeds.

| Model | Seen Objects | | | | Unseen Objects | | | |
|---|---|---|---|---|---|---|---|---|
| | PSNR ↑ | SSIM ↑ | LPIPS ↓ | FID ↓ | PSNR ↑ | SSIM ↑ | LPIPS ↓ | FID ↓ |
| *Task: Translation* | | | | | | | | |
| Chained | 13.70 | 0.309 | 0.485 | 0.94 | 14.13 | 0.326 | 0.467 | 0.97 |
| 3DIT | 15.21 | 0.300 | 0.472 | 0.24 | 15.20 | 0.292 | 0.477 | 0.25 |
| **Ours** | **20.58**$_{\pm0.62}$ | **0.439**$_{\pm0.013}$ | **0.273**$_{\pm0.008}$ | **0.15**$_{\pm0.004}$ | **20.13**$_{\pm0.81}$ | **0.429**$_{\pm0.016}$ | **0.274**$_{\pm0.008}$ | **0.16**$_{\pm0.006}$ |
| *Task: Rotation* | | | | | | | | |
| Chained | 13.18 | 0.269 | 0.540 | 1.00 | 12.85 | 0.270 | 0.538 | 1.69 |
| 3DIT | 16.86 | 0.382 | 0.429 | 0.25 | 16.28 | 0.366 | 0.447 | 0.24 |
| **Ours** | **18.52**$_{\pm0.35}$ | **0.391**$_{\pm0.012}$ | **0.354**$_{\pm0.006}$ | **0.14**$_{\pm0.002}$ | **18.39**$_{\pm0.57}$ | **0.377**$_{\pm0.016}$ | **0.365**$_{\pm0.009}$ | **0.15**$_{\pm0.008}$ |
| *Task: Removal* | | | | | | | | |
| Chained | 12.49 | 0.383 | 0.465 | 0.80 | 12.12 | 0.379 | 0.459 | 1.05 |
| 3DIT | 24.98 | 0.585 | 0.249 | 0.24 | 24.66 | 0.568 | 0.260 | 0.24 |
| **Ours** | **28.86**$_{\pm0.88}$ | **0.616**$_{\pm0.015}$ | **0.167**$_{\pm0.010}$ | **0.14**$_{\pm0.007}$ | **28.44**$_{\pm0.91}$ | **0.613**$_{\pm0.016}$ | **0.169**$_{\pm0.010}$ | **0.15**$_{\pm0.005}$ |

Table 2: **Multi-object editing results on MOVi-E, Objectron, and Waymo Open**. We compute metrics on the entire image and inside the object bounding boxes. Ours are averaged over 3 seeds.

| Model | Image-Level | | | | Object-Level | | | |
|---|---|---|---|---|---|---|---|---|
| | PSNR ↑ | SSIM ↑ | LPIPS ↓ | FID ↓ | PSNR ↑ | SSIM ↑ | LPIPS ↓ | DINO ↑ |
| *Dataset: MOVi-E* | | | | | | | | |
| Chained | 14.46 | 0.409 | 0.481 | 4.46 | 15.06 | 0.303 | 0.436 | 0.554 |
| 3DIT | 14.33 | 0.385 | 0.671 | 5.89 | 12.41 | 0.214 | 0.663 | 0.336 |
| **Ours** | **22.03**$_{\pm0.95}$ | **0.594**$_{\pm0.015}$ | **0.277**$_{\pm0.007}$ | **2.20**$_{\pm0.024}$ | **20.05**$_{\pm0.43}$ | **0.547**$_{\pm0.009}$ | **0.289**$_{\pm0.006}$ | **0.738**$_{\pm0.017}$ |
| *Dataset: Objectron* | | | | | | | | |
| Chained | 11.23 | 0.262 | 0.586 | 2.03 | 11.24 | 0.171 | 0.415 | 0.555 |
| 3DIT | 11.69 | 0.281 | 0.559 | 1.99 | 11.30 | 0.150 | 0.444 | 0.547 |
| **Ours** | **14.83**$_{\pm0.45}$ | **0.348**$_{\pm0.012}$ | **0.446**$_{\pm0.021}$ | **0.55**$_{\pm0.011}$ | **16.41**$_{\pm0.33}$ | **0.477**$_{\pm0.009}$ | **0.233**$_{\pm0.008}$ | **0.790**$_{\pm0.015}$ |
| *Dataset: Waymo Open* | | | | | | | | |
| Chained | 18.28 | **0.501** | 0.454 | 2.88 | 16.20 | 0.310 | 0.383 | 0.596 |
| 3DIT | 17.32 | 0.421 | 0.474 | 3.32 | 14.41 | 0.174 | 0.537 | 0.449 |
| **Ours** | **18.71**$_{\pm0.50}$ | 0.494$_{\pm0.018}$ | **0.404**$_{\pm0.010}$ | **1.64**$_{\pm0.003}$ | **17.67**$_{\pm0.30}$ | **0.343**$_{\pm0.011}$ | **0.348**$_{\pm0.013}$ | **0.653**$_{\pm0.019}$ |

# B Additional Experimental Results

## B.1 Full Benchmark Results

We present full quantitative results on OBJect in Tab. 1, and on MOVi-E, Objectron, and Waymo Open in Tab. 2. Compared to the main paper, we report additional FID metrics and results on the unseen object subset for OBJect, while for the other three datasets, we report additional FID and DINO feature similarity metrics, plus results computed over the entire image (Image-Level). Overall, we observe similar trends as in the main paper, where our Neural Assets model significantly outperforms baselines across all datasets. In Fig. 11, we show additional qualitative comparisons.

## B.2 Full Ablation Results

We present all quantitative results of our ablation studies on Objectron (Sec. 4.4) in Tab. 3, Tab. 4, and Tab. 5. We observe similar trends on all metrics at both image- and object-level.

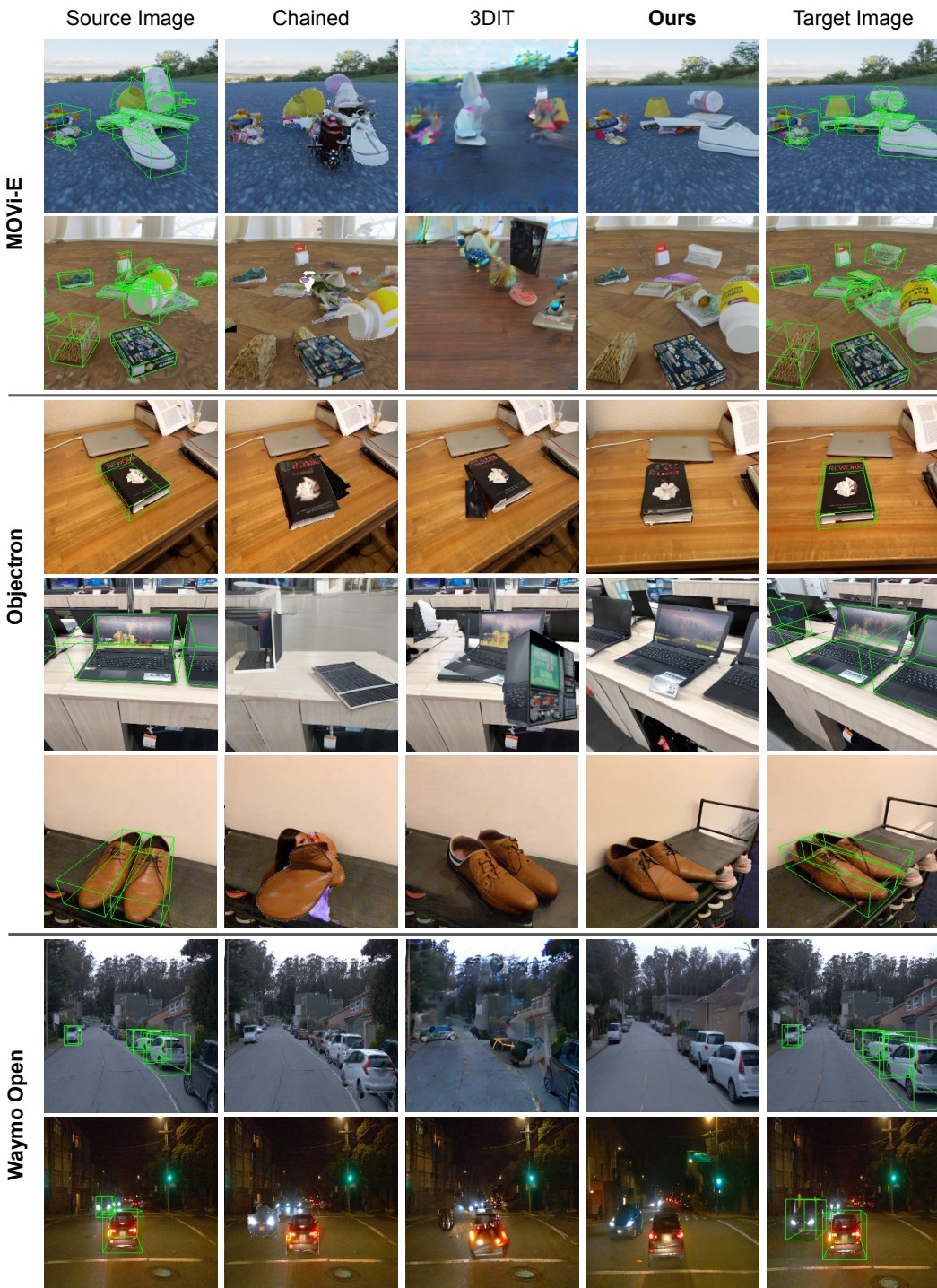

Figure 11: **More qualitative results on MOVi-E, Objectron, and Waymo Open**.

## B.3  Controllable Scene Generation

In Fig. 12 and Fig. 13, we show controllable scene generation results on Objectron. Objectron videos only have global camera movement, while the objects are static. Still, our Neural Assets model learns disentangled foreground and background representations. As can be seen from the results, we can rotate the foreground objects while keeping the background fixed, or swap background between

Table 3: **Ablation of image encoders on Objectron**. FT-DINO stands for fine-tuning DINO ViT.

| Encoder | Image-Level | | | | Object-Level | | | |
|---|---|---|---|---|---|---|---|---|
| | PSNR ↑ | SSIM ↑ | LPIPS ↓ | FID ↓ | PSNR ↑ | SSIM ↑ | LPIPS ↓ | DINO ↑ |
| CLIP | 12.95 | 0.309 | 0.532 | 0.66 | 14.13 | 0.339 | 0.333 | 0.709 |
| MAE | 13.64 | 0.317 | 0.501 | 0.59 | 14.93 | 0.369 | 0.296 | 0.735 |
| DINO | 13.75 | 0.325 | 0.498 | 0.57 | 15.03 | 0.388 | 0.296 | 0.747 |
| **FT-FINO** | **14.83** | **0.348** | **0.446** | **0.55** | **16.41** | **0.477** | **0.233** | **0.790** |

Table 4: **Ablation of background modeling on Objectron**. No-BG means not doing background modeling at all, while No-Pose stands for not using the relative camera pose between two frames.

| Background | Image-Level | | | | Object-Level | | | |
|---|---|---|---|---|---|---|---|---|
| | PSNR ↑ | SSIM ↑ | LPIPS ↓ | FID ↓ | PSNR ↑ | SSIM ↑ | LPIPS ↓ | DINO ↑ |
| No-BG | 12.98 | 0.308 | 0.513 | 1.18 | 14.49 | 0.394 | 0.297 | 0.712 |
| No-Pose | 13.71 | 0.326 | 0.496 | 0.72 | 15.39 | 0.423 | 0.273 | 0.751 |
| **Ours** | **14.83** | **0.348** | **0.446** | **0.55** | **16.41** | **0.477** | **0.233** | **0.790** |

Table 5: **Ablation of training data on Objectron**. Single and Paired refer to training on one image or source-target pairs. No-PE means removing the positional encoding in the ViT image encoder.

| Training Data | Image-Level | | | | Object-Level | | | |
|---|---|---|---|---|---|---|---|---|
| | PSNR ↑ | SSIM ↑ | LPIPS ↓ | FID ↓ | PSNR ↑ | SSIM ↑ | LPIPS ↓ | DINO ↑ |
| Single | 12.63 | 0.298 | 0.544 | 1.07 | 13.41 | 0.259 | 0.381 | 0.651 |
| Single (No-PE) | 13.74 | 0.323 | 0.503 | 1.01 | 14.51 | 0.315 | 0.337 | 0.683 |
| **Paired** | **14.83** | **0.348** | **0.446** | **0.55** | **16.41** | **0.477** | **0.233** | **0.790** |

scenes. Importantly, our model inpaints the masked background regions not occupied by the novel object, and renders realistic shadows around the object, which is far beyond simple pixel copying.

### B.4 Ablation on 3D Pose Representations

In Fig. 14, we visualize the object pose representation we use. Given a 3D bounding box of an object, we project its four corners to the image space, and concatenate their 2D coordinates and depth values to obtain a 12-D pose vector. The 2D projected points resemble a local coordinate frame for the object, specifying its position, rotation, and scale. On the other hand, the depth is useful for determining the occlusion of objects.

There are alternative ways to represent the object pose, e.g., the coordinate of the 3D box center C with its size and rotation which is commonly used in 3D object detection [57]. These representations achieve similar results on MOVi-E and Objectron. However, their learned rotation controllability is significantly worse than our representation on Waymo. This is because most of the cars on Waymo are not rotated (turn left / right), leading to very few training data on object rotation. If we directly input the rotation angle to the model, it tends to ignore it. In contrast, due to prospective projection, the projected local coordinate frame of unrotated cars still look "rotated" when they are not strictly in front of the ego vehicle. This provides much more training signal to learn the rotation of objects.

## C Background on Stable Diffusion

Diffusion model [44, 94] is a class of generative models that learns to generate samples by iteratively denoising from a standard Gaussian distribution. It consists of a denoiser $\epsilon_\theta$, usually implemented as a U-Net [80], which predicts the noise $\epsilon$ added to the data $x$. Instead of denoising raw pixels, Stable Diffusion introduces a VAE [54] tokenizer to map images to low-dimensional latent code $z$ and applies the denoiser on it. In addition, the denoiser is conditioned on text and thus supports text-to-image generation. In this work, we simply replace the text embeddings with Neural Assets $a_i$ and fine-tune the model to support appearance and pose control of 3D objects.

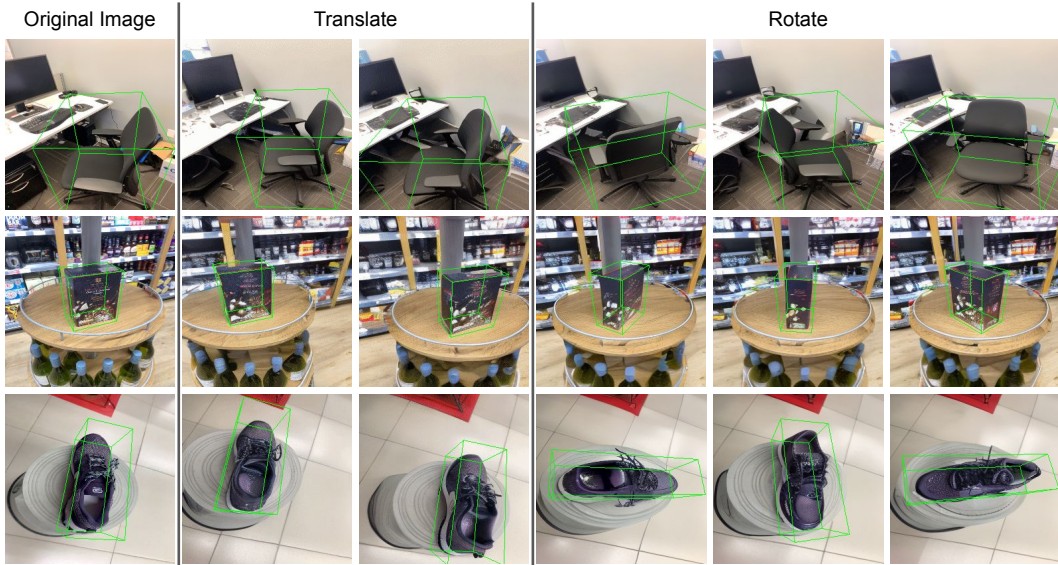

Figure 12: **Object translation and rotation results** on Objectron. Although there is only camera movement on this dataset (i.e., objects never move), the model still learns to disentangle the object pose and the camera pose. As shown in the object rotation results, the background stays fixed. See our project page for videos and additional object rescaling results.

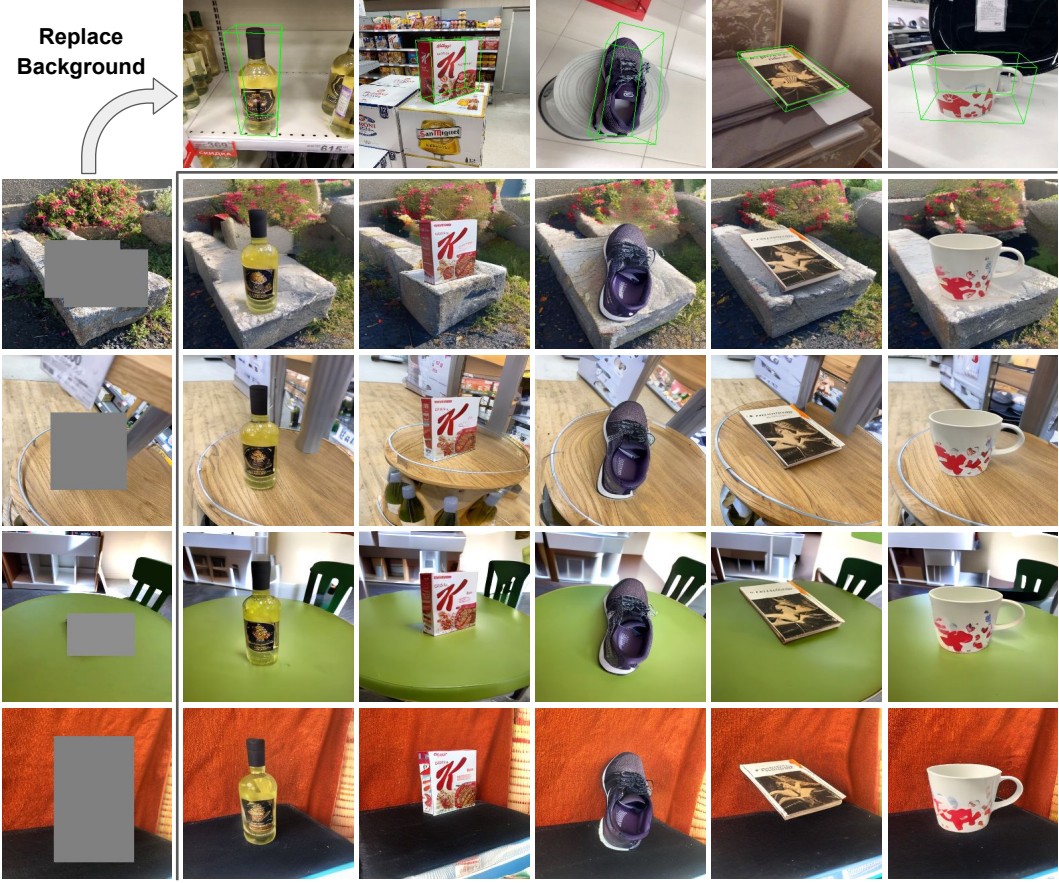

Figure 13: **Transfer backgrounds between scenes** by replacing the background token on Objectron. Note how the global camera viewpoint is adjusted to fit the foreground object. In addition, the generator is able to synthetic lighting effects such as shadows on the surfaces.

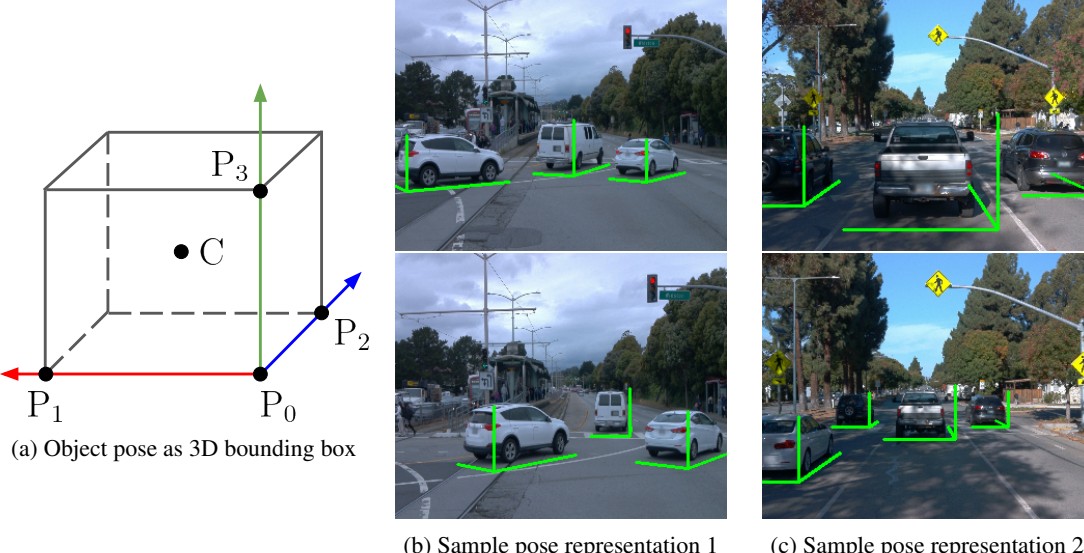

(a) Object pose as 3D bounding box

(b) Sample pose representation 1

(c) Sample pose representation 2

Figure 14: **Illustration of our object pose representation (a) and two examples (b)**. We project four corners $P_0, P_1, P_2, P_3$ of a 3D bounding box to the 2D image plane and concatenate them to obtain the pose token. The projected four corners form a local coordinate system of the object.

## D  Broader Impacts

Controllable visual generation is an important task in computer vision. Neural Assets equip generative models with an intuitive interface to control their behaviors, which enables more interpretable AI algorithms and may potentially benefit other fields such as computer graphics and robotics. We believe this work will benefit the whole research community and the society.

**Potential negative societal impacts.** Since we fine-tune large-scale pre-trained generative models in our pipeline, we inherit limitations of these base models, such as dataset selection bias. Such bias might be problematic when human subjects are involved, though our current approach is only capable of rigid object control and does not consider humans as an "asset" yet. Further study on how such bias affects model performance is required for mitigating negative societal impacts that could arise from this work [35].

## E  Funding Disclosure

This work was carried out at Google. Igor Gilitschenski contributed to the project in an advisory capacity.

