# OpenReview forum: "Neural Assets: 3D-Aware Multi-Object Scene Synthesis with Image Diffusion Models"
_NeurIPS.cc/2024/Conference — NeurIPS 2024 spotlight_

### Official Review · Reviewer_jimW · 2024-06-30

**Soundness:** 3
**Presentation:** 4
**Contribution:** 3
**Rating:** 7
**Confidence:** 4

**Summary:**

This paper presents a novel method for 3D-aware editing of multi-object images. Given the original image and 2D bounding boxes to specify the objects to be edited, and the 3D pose as the target of the editing, the proposed method encodes the object appearance feature and their pose features, which are then taken as conditions to generate the edited images. The proposed approach disentangles the appearances and poses of the objects, and enables flexible and better control of the 3D pose of each object in the image.

**Strengths:**

The strengths of this paper include:
(1)It proposes a novel approach to disentangle the appearance feature and pose feature of each object in the multi-object image.
(2)The proposed approach outperforms the STOA 3D-aware image editing methods, especially in the multi-object setting.
(3)It enables object editing and controllable scene generation applications.

Overall, this paper presents a practical solution for the 3D-aware multi-object image editing and achieves the best performance. The network design is quite straightforward, but it indeed makes the best use of the pre-trained models and the large training set.

**Weaknesses:**

Basically, this paper provides an excellent 3D-aware image editing model for multi-object images benefiting from the pre-trained image generation model and the large training dataset. The idea of disentangling the appearance and pose features of objects is the key for multi-object image editing. However, since there have been existing works on this topic, such as BlobGAN-3D, it is essential to add an in-depth discussion to validate the superiority of the proposed network architecture.

[1] BlobGAN-3D: A Spatially-Disentangled 3D-Aware Generative Model for Indoor Scenes

**Questions:**

Line 244 claims that the proposed method is not pre-trained on multi-view rendering of 3D assets. It is actually trained on the different frames of videos, which is more appropriate for the multi-object image setting. This claim is true but I would not take it as an advantage of the proposed method.

**Limitations:**

This paper includes the discussions of limitations and broader impacts. However, it's better to present some failure cases of the proposed approach to better understand its performance.

---

> ### Author Rebuttal · Authors · 2024-08-03
>
> We thank the reviewer for the positive feedback and valuable comments.
>
> **1. Discussion about prior work BlobGAN-3D [1].**
>
> A: Thanks for bringing up this related work. We will include this paper and add the following discussions in the camera-ready version of the paper:
>
> > There have been prior works learning disentangled appearance and pose representations for multi-object image editing [1, 68, 105]. However, they are based on the Generative Adversarial Networks (GANs) framework, which are pre-trained on smaller datasets and have less expressive latent space compared to recent diffusion models. In contrast, we build upon large-scale pre-trained image diffusion models, enabling editing of complex real-world scenes.
>
> [1] Wang, Qian, et al. "BlobGAN-3D: A spatially-disentangled 3D-aware generative model for indoor scenes." arXiv. 2023.
>
>
> **2. The claim of not pre-trained on multi-view images as an advantage.**
>
> A: Thanks for pointing this out. We will remove this statement in the final version of the paper.
>
>
> **3. Failure case of the proposed method.**
>
> A: Thanks for this great suggestion. We have already included failure case analysis in the website (see link in the paper or the supplementary material). We will add the following discussions in the final version of the paper:
>
> > Failure case analysis: (1) One main failure case of our model is symmetry ambiguity. As can be seen from the rotation results (Fig. 2 (a) in the uploaded PDF), the handle of the cup gets flipped when it rotates by 180 degree. (2) Another failure case that only happens on Objectron is the entanglement of global camera motion and local object movement (Fig. 2 (b) in the uploaded PDF). This is because Objectron videos only contain camera motion while objects always stay static. Both issues will likely be resolved if we train our model on larger-scale datasets with more diverse object and camera motion.

---

> > ### Comment · Reviewer_jimW · 2024-08-09
> >
> > Thank you for the response. It addresses my questions well. I would maintain my rating as "accept".

---

### Official Review · Reviewer_W8vg · 2024-07-08

**Soundness:** 3
**Presentation:** 4
**Contribution:** 3
**Rating:** 7
**Confidence:** 4

**Summary:**

This paper considers controlling the 3d poses of different objects in an image generated by a diffusion model. By conditioning the diffusion model on a sequence of per-object appearance and pose tokens instead of text tokens it is possible to finely control the object poses, and furthermore to e.g. transfer objects between scenes. The diffusion model is trained using videos with 3d bounding box annotations and correspondences between different frames, and it is shown experimentally to generate realistic images and to outperform related methods.

**Strengths:**

- The approach to replace text tokens with a sequence of per-object appearance and pose tokens is very clean and elegant.
- The results are impressive and the method achieves state-of-the-art results for the problem. The closest related work is 3DIT and the authors compare to and outperform that method on the tested datasets.
- While previous work mainly relied on synthetic data (e.g. 3DIT which introduces the OBJect dataset/task from objaverse assets) the authors propose to use 3d bounding boxes from real-world monocular videos as supervision, which has consistent appearance but different poses, which fits well for the task and can be used to generate training data.
- Qualitatively, the generated images generally look very good, especially for Waymo Open. E.g. in fig. 8 there is realistic lighting dependent on the background, so the generated objects adapt well to the rest of the scene. Also, for the scaling examples one can see that the shadows around the object mostly scale accordingly.

**Weaknesses:**

- There seems to be some issues with the modelling of the background for the Objectron dataset, though this is not an issue with Waymo Open. Failure cases are e.g. that rotations and scaling of objects also rotate and scale the background around the object, even outside the object bounding boxes. Did the authors try e.g. any alternative ways of modelling the background to fix this?
- This is not a major weakness, but there is no measure of 3d consistency, which is common in previous work, e.g.  zero-1-to-3 and follow up works often measure 3d reconstruction errors on e.g. Google Scanned Objects (GSO). It would be straight-forward to evaluate the 3d consistency of the proposed method on this dataset, and compare to e.g. zero-1-to-3. It should be mentioned though that this would not reflect the full strength of the proposed method since it can handle multiple objects and diverse backgrounds which is not captured when evaluating on that dataset.

**Questions:**

Please see weaknesses

Can the authors clarify if code and models will be released? It is mentioned that the authors “will consider” releasing code upon acceptance (line 844). Will any trained models be released? Will code to replicate the training be released?

**Limitations:**

This is adequately addressed

---

> ### Author Rebuttal · Authors · 2024-08-03
>
> We thank the reviewer for the positive feedback and valuable comments.
>
> **1. Background modeling issue on Objectron. Any attempts to fix it?**
>
> A: Objectron videos only have camera movement, while objects remain static throughout the video. Due to this data issue, the global camera motion and the local object motion are entangled, leading to background issues in the translation and rescaling results.
>
> We have tried different foreground object pose representations (see Appendix B.4) and different camera parametrizations (relative vs absolute camera pose) which serve as background pose, but none of them can fix this issue.
>
> This issue will likely disappear if we train a model on diverse data where both camera and object movement are observed, as shown in the Waymo results. Due to the lack of labeled datasets we have not pursued larger-scale training in this submission, but it should be possible in the near future thanks to recent general-purpose 3D object pose estimators. See our general response for detailed discussions on this.
>
>
> **2. Evaluation of 3D consistency on GSO.**
>
> A: Thanks for the suggestion. As the reviewer correctly pointed out, our work mainly focuses on multi-object generation, with results on real-world scenes with complex backgrounds. We compare with and outperform Zero-1-to-3 (the “Chained” baseline) in terms of novel-view synthesis. We believe the evaluations presented in the paper are sufficient to show the effectiveness of Neural Assets.
>
> On the other hand, we do not optimize for 3D consistency – this would require joint denoising of multiple views like in recent SOTA works such as CAT3D [1]. In contrast, we generate object edits/views completely independently, resulting in unavoidable inconsistencies. It is an interesting future direction to explore scene-level 3D reconstruction once we have more diverse training data.
>
> [1] Gao, Ruiqi, et al. "Cat3d: Create anything in 3d with multi-view diffusion models." arXiv. 2024.
>
>
> **3. Code release.**
>
> A: We are planning to release the inference code for Neural Assets which would allow users to attach their own training loop and dataset pipeline. It will come with an example data loading pipeline on MOVi-E and a training step implementation based on Diffusers Stable DIffusion.

---

> > ### Comment · Reviewer_W8vg · 2024-08-12
> >
> > I thank the authors for their answers. I keep my score as 7 (accept).

---

### Official Review · Reviewer_bNz4 · 2024-07-12

**Soundness:** 3
**Presentation:** 3
**Contribution:** 3
**Rating:** 7
**Confidence:** 5

**Summary:**

The paper introduces an object-centric representation for multi-object 3D pose control in image diffusion models. Instead of text token sequences, it utilizes Neural Assets, which are per-object representations learned by pooling visual features from reference images and reconstructing objects in target frames. This disentangles appearance and pose features by fine-tuning a pre-trained text-to-image diffusion model while conditioning on target frame poses, enabling fine-grained 3D pose and placement control in scenes. The results are demonstrated on both synthetic and real-world datasets in multi-object editing tasks.

**Strengths:**

1. Object-centric representation: the proposed approach introduces a novel object representation to tackle conditional image generation, which disentangles 3D appearance and 3D pose. This representation provides a new solution to tackle similar tasks in controllable 2D image / 3D scene editing tasks.
2. Experimental Results: The experiments demonstrate the effectiveness of the proposed Neural Assets in achieving fine-grained control over individual object appearances and poses within a scene. Its application is shown in both synthetic and real-world scenes, supporting tasks like background swapping and object transfers across different scenes.
3. The paper is overall well written, providing good motivations and comparison with related work. Some concerns in the method section could be further clarified (see below).

**Weaknesses:**

I value the performance the proposed model presented in manipulating objects while keeping the rest unchanged. I spent quite some time trying to comprehend how the current design leads to its performance, and still have the following questions, which might be helpful if they are discussed in the paper:
[-] The paper utilizes projection and depth of the absolute object poses in the pose embedding. This is kind of confusing as the it requires the model to encode how the object should look like in the canonical space. If this is the case, the model should be able to perform per-object reconstruction with the object-centric representation through images under different poses and even maybe object pose estimation. A common practice is to use the relative pose change, as the authors discussed in the background modeling.
[-] The authors leverage SD as base model, but one significant difference is that the proposed model put the learning bottleneck to the conditional module, compared to Zero-1-to-3 or ControlNet. All the useful features are embedded in the conditions now, including how the images should look like. In original SD, the conditional module is relatively light-weight. I'm wondering if there are limitations for the current design to capture the complex conditions and recover the input images.
[-] One related concern is that from the results, it seems the reconstruction results are also decent. I'm wondering what role is the randomness of the noise play in the current model. Will the model output different images if the seed changes?
[-] How should the model comprehend the objects in the target images that do not exist in the src images, especially during training?

Some other concerns:
[-] The model may lack a total understanding of the scene, especially the interactions between objects, e.g.,  when one object is supported by another or when the 3D bounding boxes of two objects collide with each other. How will the model react in such scenarios?
[-] The proposed model faces limitations in fully manipulating 3D object and their relationships as naturally as humans do. The requirement of 2D/3D bounding box pose challenges to the detection models and the manipulation as projection points is not intuitive as humans who commonly use language.

**Questions:**

See above. Overall, I think the paper offers a new perspective for object-centric learning, especially under the image generation settings. I still have some concerns and will consider raising my rating if the rebuttal resolves them.

**Limitations:**

Yes.

---

> ### Author Rebuttal · Authors · 2024-08-03
>
> We thank the reviewer for the constructive comments. We are glad to see the positive assessment of our paper, and will include the below discussions in the final version of the paper.
>
>
> **1. The use of absolute object pose for pose tokens.**
>
> A: This is a great question. As discussed in Sec. 3.1, a 3D asset in traditional graphics pipelines is often represented by its canonical 3D shape and its pose. Therefore, it is actually our goal to encode the object in its canonical space in the appearance token of a Neural Asset. As shown in the experiments, our model is indeed able to synthesize an object under different poses.
>
> In our preliminary experiments, we tried using relative pose changes to encode object pose tokens. However, it shows worse results compared to absolute poses. One potential cause is the use of the DINO encoder. Due to its pre-training strategy (self-distillation), DINO is encouraged to extract visual features that are invariant to image transformations (e.g., resize, translate). This objective aligns with our goal of extracting object appearance in its canonical space regardless of the observed pose.
>
>
> **2. Using only the conditional module (cross-attention) for Neural Assets conditioning.**
>
> A: The main reason we use cross-attention is that it allows us to condition the generator on individual Neural Assets which are vector-based representations, facilitating scene decomposition. It is unclear how we can apply dense conditioning such as concatenation (Zero-1-to-3) and addition (ControlNet) while still keeping the object-centric property of our method.
>
> While we do not use pixel-aligned conditioning, full fine-tuning of the visual encoder and diffusion model can greatly improve the generation result (see ablations in Fig. 9 (a)). We agree that there are still artifacts in local visual details. We tried increasing the RoIAlign size of each object ($K$ for the appearance tokens) but it did not help much. Using a stronger base model (e.g. Stable Diffusion XL) or image encoder (e.g. DINO v2) might solve this problem, which we leave for future works.
>
>
> **3. Effect of random seeds in the generation results.**
>
> A: As we condition the generator on object and background representations, we should expect small variations in the generated images. Yet, since novel-view synthesis is an ill-defined problem, the model needs to hallucinate new content in regions unobserved in the source image. Fig. 1 of the uploaded PDF shows the generation results from our model under three random seeds. The global scene layout and object geometry are identical among different seeds. In addition, our model synthesizes diverse but plausible variations in the local details of objects and backgrounds.
>
>
> **4. New objects in the target image that do not exist in the source image.**
>
> A: New objects will only have a valid pose token, while the appearance token is set to zero. The model is encouraged to hallucinate it. In fact, in order to apply classifier-free guidance (CFG), we intentionally set the appearance tokens to zero with a probability of 10% during training. Thus, the new objects will be treated similarly by the model without harming the performance.
>
>
> **5. The ability to model object interactions in the scene.**
>
> A: We leverage cross-attention to inject Neural Assets to the latent features. It is true that object tokens do not interact with each other directly in cross-attention. However, the following self-attention should learn the object interactions and generate an image with coherent content. Our hypothesis is that object interaction does not need to be architecturally hardcoded into the conditioning branch.
>
> As shown in Fig. 6 and Fig. 7 of the main paper, our model handles object occlusions correctly. As shown in Fig. 8 of the main paper, our model adapts objects to new backgrounds, e.g., the car lights are turned on at night. These results prove that our model understands object-object and object-background interactions.
>
> Regarding object collision, the Waymo results on our website (see link in the paper or the supplementary material) show a few cases (Rotation on row 1 & 4). The colliding objects blend into each other, while the other parts of the image look normal. We argue that as a conditional generator, our model is tasked to follow the input object poses. If the 3D bounding boxes are physically wrong such as collision, the model will just generate implausible images instead of correcting it.
>
>
> **6. Limitation of using 3D bounding boxes for control & Extension to language control.**
>
> A: Please see our general response about the discussion of using datasets with 3D bounding box annotations. With recent general-purpose 3D object pose estimators [1], we can build a large-scale training dataset with more diverse scenes to learn Neural Assets of general objects.
>
> [1] Krishnan, Akshay, et al. "OmniNOCS: A unified NOCS dataset and model for 3D lifting of 2D objects." ECCV. 2024.
>
> We agree that language-based control is more intuitive. But it is harder to achieve precise spatial control of objects compared to using bounding boxes. We thus leave this as an interesting future direction.

---

> > ### Comment · Reviewer_bNz4 · 2024-08-08
> > **Post Rebuttal**
> >
> > I thank the authors for their responses.  They address my questions about the model design and qualitative results, and thus I have raised my score accordingly.

---

### Official Review · Reviewer_h1dc · 2024-07-14

**Soundness:** 4
**Presentation:** 4
**Contribution:** 3
**Rating:** 6
**Confidence:** 4

**Summary:**

This paper addresses the task of multi-object pose and scale control in image generation and editing using diffusion models. The authors introduce an object-centric representation with a disentangled pose and appearance called Neural Asset. The neural asset for each object in an image is estimated using pose and appearance encoders. The authors exploit the naturally occurring pose and appearance variations in training video datasets to train such encoders. To generate an image based on neural assets, the authors fine-tune an image diffusion model conditioned on the sequence of neural assets as its condition. The authors evaluate and compare their method with existing baselines, showcasing the ability of their method for pose and scale control, object removal, and background change.

**Strengths:**

- The paper is well-written and easy to follow
- The proposed method is reasonable and of sufficient novelty
- The provided results show the efficacy of the proposed approach for modeling multiple objects in images and controlling their pose and scale
- The experiments are sufficient and contain ablation study on different design choices

**Weaknesses:**

- Based on the provided results, the background has small changes when an object in the image is edited even in the areas far from the edited object
- The proposed method is mainly limited to the domain of the training dataset as opposed to the recent training-free diffusion-based image editing methods

**Questions:**

See the weaknesses

**Limitations:**

sufficiently discussed

---

> ### Author Rebuttal · Authors · 2024-08-03
>
> We thank the reviewer for their detailed feedback and encouraging comments.
>
> **1. Small background changes in areas far from the edited objects.**
>
> A: We encode object appearance tokens by applying RoIAlign on the image feature map. Even if we use the paired frame training strategy for feature disentanglement, there is still background information leak to the foreground object representations. As a result, editing an object might lead to small changes in an irrelevant background region.
>
> This is not specific to our work: small background variation is a common issue in 3D-aware image editing methods (e.g., see results on the website of Diffusion Handles [69] and LooseControl [6]), which might be resolved with more powerful base models and larger-scale training data. We thus leave it for future work.
>
>
> **2. Limited application domain of Neural Assets compared to training-free methods.**
>
> A: Please see our general response about the discussion of using datasets with 3D bounding box annotations. With recent general-purpose 3D object pose estimators [1], we can build a large-scale training dataset with more diverse scenes to learn Neural Assets of general objects.
>
> [1] Krishnan, Akshay, et al. "OmniNOCS: A unified NOCS dataset and model for 3D lifting of 2D objects." ECCV. 2024.

---

> > ### Comment · Reviewer_h1dc · 2024-08-13
> > **Final Comment**
> >
> > I thank the authors for their response. I keep my score.

---

### Author Rebuttal · Authors · 2024-08-03

We would like to thank the reviewers for their helpful feedback and insightful comments.

We are glad that the reviewers find our paper “*well written*” (h1dc, bNz4), our Neural Assets framework “*elegant*” (W8vg) and “*novel*” (h1dc, bNz4, jimW). Also, our experiments are considered “*sufficient*” (h1dc), “*impressive*” (W8vg), and “*providing good motivations and comparison with related work*” (bNz4).

From the view of generative models, all reviewers acknowledge that our method supports multi-object pose control, enabling several editing tasks. Reviewer W8vg points out that “*the generated images generally look very good*” such as “*realistic lighting*”. Finally, reviewer jimW agrees that our model “*makes the best use of the pre-trained models and the large training set*”.

We have uploaded a PDF file that includes figures to address the reviewers’ feedback. Below we include a response to a general question raised by the reviewers. For other questions, please see our response to individual questions below each review. We will incorporate all our responses and additional results in the final version of the manuscript.

- **Limitation of using 3D bounding boxes for pose encoding** (reviewers h1dc, bNz4, and W8vg)

At the time of paper submission, there was no general-purpose 3D object pose estimator available. Therefore, our experiments are limited to datasets with 3D bounding box annotations. However, a recent work OmniNOCS [1] fills this gap by introducing a 3D pose estimator that works on both Waymo and Objectron (datasets we used in this work), and *diverse, in-the-wild Internet images* for a wide range of object classes. With recent advances in vision foundation models, we expect to see scalable 3D annotation pipelines similar to their 2D counterparts soon, which could be used to learn Neural Assets of more general objects in more diverse environments.

[1] Krishnan, Akshay, et al. "OmniNOCS: A unified NOCS dataset and model for 3D lifting of 2D objects." ECCV. 2024.

---

### Comment · Area_Chair_43uf · 2024-08-12
**Authors' Response**

Dear Reveiwer h1dc,

The authors have provided a response to the questions that you raised in your review. It would be great if you could read and acknowledge their response before 11:59 pm AOE August 13, 2024.

Best,
AC

---

### Decision · Program_Chairs · 2024-09-25

**Decision:**

Accept (spotlight)

**Comment:**

This paper proposes a method for controlled generation of compositional scenes using diffusion models with explicit pose control of the individual objects in the scene. To achieve this the authors propose a new Neural Assets representation, which disentangles the identity and the pose of the object. These representations are further appended into the text prompt provided to the diffusion model for conditioning. Additionally the authors propose a novel methodology to train their system using videos containing the objects in different poses in different frames, which further helps to disentangles the identity and expression representations. On several benchmark datasets the authors show significant improvement in performance over the state of the art and show for the first time 3D control for composes scenes instead of just single objects.

All reviewers appreciated the novelty and elegance of the proposed method and the high quality of the results achieved. They further appreciated the well-written nature of the paper. All reviewers further championed the acceptance of this work. This work represents a significant step forward in synthesizing 3D controlled scenes composes of individual objects. The AC concurs and recommends acceptance.